# Stability Considerations for Bacteriophages in Liquid Formulations Designed for Nebulization

**DOI:** 10.3390/cells12162057

**Published:** 2023-08-12

**Authors:** Rohan Flint, Daniel R. Laucirica, Hak-Kim Chan, Barbara J. Chang, Stephen M. Stick, Anthony Kicic

**Affiliations:** 1School of Biomedical Sciences, The University of Western Australia, Perth, WA 6009, Australia; rohan.flint@telethonkids.org.au; 2Wal-yan Respiratory Research Center, Telethon Kids Institute, The University of Western Australia, Perth, WA 6009, Australia; daniel.laucirica@telethonkids.org.au (D.R.L.); stephen.stick@health.wa.gov.au (S.M.S.); 3Advanced Drug Delivery Group, School of Pharmacy, University of Sydney, Sydney, NSW 2050, Australia; kim.chan@sydney.edu.au; 4The Marshall Center for Infectious Diseases Research and Training, School of Biomedical Sciences, The University of Western Australia, Perth, WA 6009, Australia; barbara.chang@uwa.edu.au; 5Department of Respiratory and Sleep Medicine, Perth Children’s Hospital, Perth, WA 6009, Australia; 6Centre for Cell Therapy and Regenerative Medicine, School of Medicine and Pharmacology, Harry Perkins Institute of Medical Research, The University of Western Australia, Perth, WA 6009, Australia; 7School of Population Health, Curtin University, Perth, WA 6102, Australia

**Keywords:** bacteriophages, antimicrobial resistance, respiratory infections, nebulization, aerosolized delivery

## Abstract

Pulmonary bacterial infections present a significant health risk to those with chronic respiratory diseases (CRDs) including cystic fibrosis (CF) and chronic-obstructive pulmonary disease (COPD). With the emergence of antimicrobial resistance (AMR), novel therapeutics are desperately needed to combat the emergence of resistant superbugs. Phage therapy is one possible alternative or adjunct to current antibiotics with activity against antimicrobial-resistant pathogens. How phages are administered will depend on the site of infection. For respiratory infections, a number of factors must be considered to deliver active phages to sites deep within the lung. The inhalation of phages via nebulization is a promising method of delivery to distal lung sites; however, it has been shown to result in a loss of phage viability. Although preliminary studies have assessed the use of nebulization for phage therapy both in vitro and in vivo, the factors that determine phage stability during nebulized delivery have yet to be characterized. This review summarizes current findings on the formulation and stability of liquid phage formulations designed for nebulization, providing insights to maximize phage stability and bactericidal activity via this delivery method.

## 1. Introduction

Pulmonary infections are one of the leading causes of morbidity and mortality worldwide [1], severely impacting individuals with chronic respiratory diseases (CRDs) such as cystic fibrosis (CF) and chronic obstructive pulmonary disease (COPD). Individuals with CRDs are prone to persistent infections with opportunistic pathogens leading to extensive inflammation and lung damage [2,3,4,5,6]. One of the most important bacteria implicated in these infections is *Pseudomonas aeruginosa*, which is a gram-negative pathogen associated with a suite of negative health impacts for individuals with CRDs [7,8]. In individuals with CF, infection with *P. aeruginosa* results in increased airway inflammation and mortality as well as an increase in lung function decline [5,6,9,10]. Within those with COPD and non-CF bronchiectasis, up to one-third of individuals may be colonized by *P. aeruginosa*, with infection being linked to an increase in lung function decline and mortality [11,12,13].

Like other bacterial pathogens, *P. aeruginosa* is capable of accumulating antimicrobial resistance (AMR) to multiple classes of antibiotics, becoming multi-drug resistant (MDR). *P. aeruginosa* is also capable of forming biofilms, further reducing antibiotic efficacy and contributing to recalcitrant and recurring infections [14,15]. The impact of respiratory antimicrobial-resistant infections is increasing globally, with lower respiratory infections accounting for over 1.5 million deaths [16]. For those with CRDs, pulmonary infection with antibiotic-resistant bacteria has been further linked to an increase in disease severity and mortality [9,17,18]. Antimicrobial-resistant infections can be exceedingly difficult and costly to treat, especially in individuals with CRDs who are persistently infected [7]. Currently, intravenous (IV) and aerosolized antibiotics are used to treat pulmonary bacterial infections including those caused by *P. aeruginosa*; however, as bacteria are becoming increasingly antimicrobial-resistant, the effectiveness of these antibiotics is diminishing [19,20]. With little investment in the discovery and development of antibiotic compounds, there is an urgent need for novel antimicrobials [21]. This is where pulmonary phage therapy may serve as a promising alternative or complementary therapy to treat antimicrobial-resistant infections in the lungs. The method of delivery for therapeutic phages can vary depending on the site of infection. Skin infections can be treated with a topical ointment containing phage [22], but the delivery of phages to deeper sites of infection such as the lungs is more challenging. While intravenous phage delivery has been used previously to treat respiratory infections, the inhalation of phage allows for the direct delivery of the phage to sites of respiratory infection, and has been proposed to be a more promising method of delivery [23].

The aerosolization of phage through nebulization and dry powder inhalers has so far shown promising success; however, many factors within the delivery process remain poorly understood. The elucidation of these factors will allow for the optimization of aerosolized phage therapy, ensuring its success as a therapeutic in the future.

## 2. Phage Therapy to Treat Antimicrobial Resistance

Bacteriophages or phages are viruses that solely infect and replicate within bacterial cells. They are among the most abundant biological entities on the planet and are ubiquitous in the environment, with an estimated 10^31^ phage particles in the biosphere [24]. They were first discovered in the 1910s independently by Frederick Twort and Felix d’Herelle, with early studies in humans indicating their effectiveness for treating bacterial infections [25,26,27,28]. The discovery of penicillin by Fleming in 1928, coupled with the lack of widespread resistance to antibiotics at the time, led to phage therapy being overlooked in Western countries as antibiotic drugs were further developed [29]. However, phage therapy has seen continued use in some Eastern European countries [29,30,31].

With the emergence of AMR, phage therapy has seen a resurging interest as a possible solution [32]. Phage therapy implements shorter treatment periods than antibiotics and has shown effectiveness against pathogens that are antibiotic resistant [33,34]. A systematic review by Liu et al. on the safety of phage therapy found that serious adverse events directly associated with phage therapy were extremely rare [35]. In contrast, the higher doses of antibiotics currently required to treat MDR infections may result in significant side effects, including the development of drug allergies as well as high levels of ototoxicity and nephrotoxicity [36,37,38].

Phages can exhibit a lysogenic life cycle, where they integrate into the host genome and can alter the phenotype of bacteria [39,40]. For a phage to be considered therapeutically useful, it must have a strictly lytic life cycle, where the phage first infects and replicates within a bacterial cell, which is followed by lysis of the bacteria and the release of infectious phage particles [41,42,43]. Due to the specificity of phage for their target bacteria, phage therapy allows for there to be no adverse effect on normal flora when compared to high-dose, broad-spectrum antibiotic treatment [44]. However, this specificity may also require phages to be screened against individual bacterial species to ensure they can lyse patient bacterial isolates. Phage therapy often involves the use of multiple phages with a broad host range against isolates within a bacterial species [45]. These phages are used together in a formulation termed a ‘phage cocktail’ that can be used to overcome potential bacterial phage-resistance mechanisms [45,46]. Most contemporary instances of phage therapy have been approved on the basis of compassionate use and have shown encouraging success in humans [47,48,49].

## 3. Phage Delivery for Respiratory Infections

### 3.1. Intravenous Delivery

Intravenous delivery is the phage delivery method most commonly used for respiratory infections in humans. Early trials of phage therapy, as well as more recent case reports, have shown IV phages to be effective for respiratory infections [27,50]. However, the IV delivery of phages has been suggested to stimulate both the innate and adaptive immune systems, leading to a reduction in phage titer [51,52,53,54]. Sokoloff and colleagues suggested that as the bloodstream is not a natural environment for phages, the peptides contained on the phage surfaces are recognized by antibodies, leading to complement activation and clearance from the body [54]. Notably, the application of phages topically to the skin or to mucosal surfaces such as the oral cavity has not been shown to result in an immunological response [55,56,57].

### 3.2. Aerosolized Delivery

Sites of infection in the lungs are not easily accessed and as such, the systemic administration of phages or antibiotics may fail to deliver a sufficient quantity to the lungs to clear an infection. In these scenarios, direct inhalation appears to be a more promising method, which can be achieved via the inhalation of a nebulized liquid or a dry powder. Aerosolized medication requires the formation of small particles capable of penetrating deep into the lung, below 5 µm in diameter [58]. Antibiotics have been delivered via aerosolization since the 1940s, and this has been shown to result in a higher concentration of drug in the lungs compared to systemic administration [59,60]. The inhalation of antibiotics including tobramycin is common for those with CRDs such as CF, particularly for treating infections with *P. aeruginosa* [61,62]. Similarly, the inhalation of phages through various methods of aerosolization seems to be a promising method for phage delivery directly to sites of infection in the lung [23]. Unlike antibiotic compounds, phages are biological entities that may be sheared and inactivated by the mechanical stresses of aerosolized delivery methods, and they must remain intact and functional after delivery in order to exert a bactericidal effect [63]. Perhaps the most important concern for phage aerosolization is the loss of phage viability through the mechanical destruction of virions during the delivery process, which is a challenge that has been observed for all forms of aerosolization [64,65,66,67,68]. In separate studies, Carrigy et al. and Astudillo et al. assessed three types of nebulization and observed a reduction in phage titer in all cases [64,65]. Additionally, various studies have observed phage titer loss in the production of inhalable dry powders [66,67,68].

#### 3.2.1. Nebulization

Nebulization involves the process of converting a liquid solution into a fine aerosol. It is considered the optimal method for the pulmonary delivery of drugs due to its ability to deliver large volumes easily and in particle sizes capable of depositing in lower airways [69,70]. There are various types of commercial nebulizers available that have been tested with phage preparations, including jet nebulizers, vibrating or static mesh, and ultrasonic nebulizers. The general principle of phage delivery with these nebulizer types is represented in Figure 1.

Jet nebulization (Figure 1A) involves a compressed gas forced through tubing to create a zone of low pressure around a nozzle. This zone of low pressure causes liquid to rise from the reservoir, creating primary and secondary droplets [71,72]. Large primary droplets may impact on the walls of the nebulizer or on the primary baffle and are recycled into the reservoir, whereas smaller droplets that avoid this recycling process form the respirable portion of generated aerosol [71,73]. Ultrasonic nebulizers (Figure 1B) generate an aerosol via the oscillation of a piezoelectric crystal at high frequencies to generate and transfer acoustic energy waves to a liquid solution, leading to aerosol production [74]. A large amount of heat is often produced during ultrasonic nebulization, which may lead to the degradation of organic proteins and phages within the nebulization buffer, and as such, ultrasonic nebulization is not recommended for highly viscous or proteinaceous solutions [74,75]. There are two subclasses of mesh nebulizers: the first is vibrating mesh nebulizers (Figure 1C), which generate an aerosol by the action of a piezoelectric element that vibrates (100–300 kHz) an aperture plate lined with conical-shaped holes [75]. As the mesh vibrates, the liquid drug solution is forced through the holes and ejects as an inhalable aerosol [75,76]. In contrast, static mesh nebulizers (Figure 1D) utilize an ultrasonic transducer in contact with a liquid drug to generate vibrations (150–250 kHz) within the drug, pushing the solution through a static mesh to generate an aerosol [77].

Mesh nebulization has been shown to generally result in a lower reduction in phage titer than other nebulizer types [64,65]; however, the mechanisms behind this are not well characterized. In their study on the anti-tuberculosis phage D29, Carrigy et al. observed a 3.7 log_10_ titer reduction from jet nebulization as opposed to a 0.4 log_10_ and 0.6 log_10_ reduction for vibrating mesh and soft mist inhalation, respectively [65]. Carrigy et al. note that while the flow rate of jet nebulized aerosol exiting the mouthpiece was 0.122 (±0.003) mL/min, the flow rate of aerosol exiting the atomizing nozzle within the device was 17.6 (±8.1) mL/min. This indicates that only a small fraction of the aerosol passing through the nozzle was exiting the device through the mouthpiece. The majority of droplets were too large for inhalation, causing them to impact on the primary baffle and interior surface of the nebulizer and then fall back into the reservoir for re-nebulization (Figure 1A). Using their mathematical model, Carrigy and colleagues discovered that phages were subjected to the stress of impaction on the baffle an average of 96 times before exiting the mouthpiece [65]. Their results were in agreement with values from the literature regarding the Collison jet nebulizer, for which 99.92% of aerosol exiting the nozzle is re-nebulized before exiting the mouthpiece [78]. Although this might appear to account for the higher titer reduction, Carrigy et al. also observed that the first 10.5% of phage exiting the mouthpiece still had a titer reduction of over 3 log_10_ despite undergoing only 1–17 nebulizer cycles [65]. The study concluded that any degree of impaction onto the primary baffle was resulting in the large reduction in titer. In contrast, other studies have found that some phages are less damaged by jet nebulization compared to vibrating mesh and that nebulizer effects on phage viability are specific to individual phages [69,70,79,80]. Sahota et al. used a cascade impactor to compare the effect of the Omron vibrating mesh nebulizer and the AeroEclipse jet nebulizer on two *P. aeruginosa* phages [70]. They found that jet nebulization resulted in a 1 log_10_ reduction in titer, while the vibrating mesh nebulizer caused a reduction of approximately 2 log_10_ [70]. Additionally, Golshahi et al. noted a similar titer reduction in a *Burkholderia cepacia* complex phage between the Pari LC star jet nebulizer and the Pari eFlow vibrating mesh nebulizer [69]. Various in vitro and in vivo studies assessing phage viability following nebulization are presented in Table 1.

The contrasting findings across these studies make it difficult to predict how a phage will be affected by nebulization, and no definitive phage characteristics have been associated with loss of activity. In addition, studies examining aerosolized phage viability may not always assess the same parameters, making their comparison challenging. Leung et al. have suggested that an increase in the length of phage tails is associated with a higher loss of phage; however, another study involving a number of other phages did not reach the same conclusion [81,82]. Both these studies have assessed these characteristics using jet nebulization but not vibrating mesh nebulization. A study by Astudillo et al. used transmission electron microscopy (TEM) to provide visual proof of phage damage from jet and mesh nebulization, and found that a significant portion of phage damage was due to tail detachment, indicating this may be a weak point for phage during mechanical stress [64]. However, phage damage via nebulization has also been observed for phages without tails, and is suspected to be a consequence of nucleic acid ejection from mechanical stresses [64,85].

Despite the challenges of phage nebulization, it remains a promising method of delivering phages directly to sites of infection in the lower airways. In addition, when compared to intravenous administration, it does not lead to high stimulation of the host immune system [52,53,54]. A recent study by Liu and colleagues assessed the use of jet nebulization to deliver phage D29 in a murine model, demonstrating the effectiveness of aerosolized phage for pulmonary delivery [86]. They observed approximately 10% of the aerosolized dose being delivered to the lungs of infected mice compared with <0.1% lung delivery for intraperitoneal injection [86]. Similarly, Semler et al. compared the reduction in bacterial load in the lungs by phage delivered by jet nebulization and intraperitoneal injection in mice [83]. They observed up to a 4 log_10_ reduction in the bacterial load in the lungs from phages delivered by nebulization as opposed to a 0.5 log_10_ reduction for phages delivered intraperitoneally [83]. 

There may be a precedent for some cases of phage therapy to utilize a combination of intravenous and aerosolized delivery, particularly when a lung infection has spread systemically. In a recent study, Prazak et al. assessed the nebulized and intravenous delivery of phages to treat methicillin-resistant *Staphylococcus aureus* (MRSA) pneumonia in rats [53]. They found that the combination of intravenous and aerosolized delivery led to a more effective eradication of bacteria than either treatment separately. They postulated this was likely due to the inability of inhaled phage to treat MRSA that escaped the lung and spread throughout the body, which was confirmed by the presence of MRSA in the spleen [53].

#### 3.2.2. Dry Powder Inhalers

Phages may also be delivered as a solid preparation generated by a freeze or spray drying process, allowing for ease of transport as well as extended storage capabilities [80,87]. Several studies have shown freeze-drying to be an effective method for stabilizing phage for long-term storage [67,88,89]. Compared to nebulizers, dry powder inhalers (DPIs) are portable and simple to use, and they do not require extensive cleaning and disinfection. During the process of freeze-drying phages, sugars such as sucrose, lactose and trehalose may provide a high level of protection to phages from freezing and drying stresses as well as for long-term storage [67,88]. The levels of log loss associated with powdered phage storage and delivery are similar to those of liquid preparations, and phage powders are capable of retaining their activity over several years in storage with minimal titer loss (<2 log_10_) [67]. Powder inhalers may cause a reduction in phage viability as the powder is dispersed through both impaction and shear forces, and much like nebulization, these forces are considered phage and formulation dependent [66,68]. Despite this, many studies have demonstrated the effectiveness of inhaled phage powders, and it remains a possible method of phage delivery for respiratory infections [90,91,92].

## 4. Phage Formulations for Aerosolization

### 4.1. Stability and Storage

An important consideration for pulmonary delivery is the stability of phage in their pre-delivery formulation, as this may in turn affect phage viability and performance upon aerosolization. Phages exhibit highly variable stability across different formulations including liquid, gels and powders, especially across different phage classes and types [93,94]. Phages often exhibit poor stability in pure water, where direct oxidation can result in capsid and tail degradation, leading to a loss of genetic material [95]. This may complicate the production of therapeutic phage preparations for human use, where ultrapure water may be required in the manufacturing process. For long-term storage, phage preparations are often contained in buffer systems including Tris, salt–magnesium (SM) and phosphate-buffered saline (PBS) [63,96,97]. Cooper et al. assessed the long-term storage stability of three separate liquid phage preparations in PBS over 6 months with no significant loss of infectious titer (≤0.5log) when stored at 4 °C or at room temperature; however, they only tested the storage stability of their cocktail components and not their nebulizer cocktail [63].

The addition of divalent ions such as Mg^2+^ and Ca^2+^ into phage buffers may also have a protective effect for the long-term storage of phage by stabilizing the phage tail; however, the effect these ions have during nebulization is not known [96,98]. Phage preparations intended for human use are commonly diluted in 0.9% medical saline before pulmonary delivery [99]. A study by Carrigy et al. assessed the suitability of this method for aerosolization and found minimal effects of saline dilution using a vibrating mesh nebulizer (≈0.5 log_10_ reduction), whereas a jet nebulizer resulted in a large titer reduction of ≈3.7 log_10_ PFU/mL [65]. However, the study concluded that the increased reduction was due to the mechanical forces of the jet nebulizer itself rather than the saline dilution step [65]. Conversely, studies by Liu et al. and Verreault et al. found that deionized water was the optimal nebulizer fluid for jet nebulization [100,101]. Both studies postulated the presence of strong ionic charge and salt concentration in both PBS and 0.9% saline were deleterious to phage stability during jet nebulization; however, this has not been extensively tested using vibrating mesh nebulization [100,101]. It is worth noting that the particle size distribution of the aerosols was not altered by the change in buffer [100,101]. If strong ionic content is indeed detrimental to phage nebulization, the addition of ions such as Ca^2+^ and Mg^2+^ to stabilize liquid phage preparations, as is commonly performed, may inadvertently result in a loss of activity. Further research is required to understand for which phages ions may be beneficial or deleterious. These findings highlight a potential challenge when creating a pharmaceutical phage preparation, as there is no guarantee a patient will have access to a specific type of nebulizer. Therefore, a pharmaceutical phage preparation may need to be trialed with multiple nebulizer types to ensure it remains therapeutically effective with different nebulizer technologies.

### 4.2. Organic Additives

Some studies have identified that the presence of proteins or organic fluid in the aerosolization medium results in a protective effect on aerosolized viruses, including phage [85,102]. This has also been reported for nebulized antibody therapy, where the addition of surfactants and increased protein concentrations have been shown to protect antibodies against aggregation and interfacial stresses [103,104]. Turgeon and colleagues directly assessed the addition of organic fluid using five tail-less phages; however, they were only able to test this using one jet nebulizer model (TSI 9302 atomizer) [85]. This was due to their use of amniotic fluid as an organic supplement, which led to clogging during tests of the vibrating mesh nebulizer (Aeroneb Lab) and the second jet nebulizer (Collison 6) [85]. Out of the five tail-less phages tested, the presence of organic fluid resulted in a protective effect for two phages, PR772 and Φ6 [85]. Their results are analogous to those from a previous study by Phillpotts et al. which suggested organic molecules in an aerosolization buffer may enhance the stability of Φ6 [105]. In their study, Phillpotts and colleagues tested the stability of Φ6 in an aerosol following jet nebulization with the Collison nebulizer [105]. The aerosol generated was kept within a rotating steel drum with samples being collected over time. The authors tested spray fluids of LB broth with either 2% skimmed cow’s milk or 2% bovine serum albumin as well as a mammalian cell culture medium Leibovitz L15 supplemented with FCS, HEPES and glutamine [105]. They observed a significantly higher survival for phages using Leibovitz L15 as the spray fluid [105]. This finding has yet to be established for tailed phages, which make up the majority of phages in nature [106]. The use of these bovine components in phage formulations may also not be translatable for clinical studies in humans.

To date, no studies have assessed the use of organic fluid in the nebulization buffer to protect phages with an ultrasonic or mesh nebulizers. Due to the heat produced during ultrasonic nebulization, they are not recommended for viscous solutions and would likely degrade organic proteins and potentially phages in the nebulization buffer [72]. Although mesh nebulization is considered less damaging to phages, their use with organic proteins is questionable [64,107]. With vibrating mesh nebulizers, a small increase in viscosity may lead to an increase in the fine particle fraction (FPF); however, as the viscosity increases beyond a certain limit, aerosol generation may cease entirely for both vibrating mesh and ultrasonic nebulization [107,108]. Therefore, the use of organic fluid to protect phages may only be suitable for jet nebulization, which may be beneficial to counteract the higher loss of phage titer commonly reported with this method. Furthermore, some air-jet nebulizers have shown the capacity to generate aerosols from highly viscous solutions [72,108].

### 4.3. Dosage

Following preparation, storage, and delivery processes, phages must be delivered in a high enough titer to the site of infection to result in sufficient bacterial clearance. In mice, it is generally accepted that the minimum effective therapeutic dose for phage therapy is 1 × 10^7^ PFU [69]. Although the required dose for effective therapy in humans is not known, many researchers report a phage quantity of approximately 10^8^ PFU is required to maximize bacterial clearance [30,109,110,111]. However, there are currently no consistent delivery and dosing regiments for nebulized phage therapy, and the optimization of the required dosage may depend on multiple factors, including the phage/bacteria ratio and dosing frequency for treatment as well as the type and model of nebulizer used for delivery. In addition, any underlying conditions in a patient may influence the efficacy of phage delivery [112]. As many characteristics of aerosolized phage therapy are yet to be elucidated, the optimal method of delivery of phage for a patient will depend on their individual case. This is further complicated by the highly specific and variable nature of phages to different aerosolized delivery methods [79,80].

## 5. Aerosolized Phage Therapy in Humans

The use of aerosolized phage therapy has been shown to be a promising method for the treatment of MDR infections in humans [99,113,114]. Human case reports have thus far been primarily limited to compassionate use cases, often involving a combination of inhalation and IV delivery. In the case of a 52-year-old individual with an MDR *Acinetobacter baumannii* infection, Rao and colleagues assessed IV delivery alone as well as a combination of IV and nebulization [115]. Interestingly, they observed greater clinical improvement with combination therapy when compared with IV delivery alone and noted that nebulization may be a more efficacious method for delivering phages to infection sites in the airways [115]. Lebeaux et al. reported aerosolized phage therapy in a 12-year-old lung-transplanted CF individual with a pandrug-resistant *Achromobacter xylosoxidans* lung infection [113]. In this case, complete clearance of the pathogen was only observed almost two years after phage therapy ceased. It is uncertain whether the patient’s condition would have improved without phage therapy; however, the authors noted that multiple rounds of antibiotic treatment and oxygen therapy did not result in any lung function improvement [113]. After two rounds of phage therapy, the patient’s respiratory condition slowly improved across the two-year period until bronchoalveolar lavage (BAL) cultures were negative for *A. xylosoxidans* [113].

It can be difficult to determine the exact clinical efficacy of aerosolized phages with respect to bacterial clearance, as the primary outcome for compassionate use cases is generally a swift clinical improvement and the reduction in pathogen load is not always quantitatively measured. However, in a recent case report of nebulized phage therapy by Köhler et al., sputum samples collected 8 h post-nebulization showed a maximum of 5 log_10_ PFU despite delivery of over 9 log_10_ phage [116]. Although the authors do not mention the type of nebulizer used, this indicates the reduction in phage titer commonly observed following aerosolized delivery. Despite this, the authors suggested that the increase in phage number alongside a reduction in bacterial load indicated that inpatient phage replication following inhalation had occurred [116]. Table 2 presents various case studies of phage therapy in humans using nebulization. The major outcomes of these case reports indicate that aerosolized phage delivery generally results in a significant clinical improvement for patients with the potential to result in a complete eradication of MDR respiratory pathogens.

## 6. Considerations for Future Development of Aerosolized Phage Therapy

The primary aim of aerosolized phage therapy is to ensure the delivery of an efficacious dose of viable phages to a site of respiratory infection. The various mechanisms in which the titer or number of viable phages may be reduced during the formation and delivery of aerosolized phages are presented in Figure 2. Damage to phages during manufacture and storage may result in an unacceptable level of titer reduction before delivery (Figure 2A). The optimization of phage stability in storage through the addition of stabilizing additives and adjuvants will ensure there is no significant loss of titer before aerosolization. The titer reduction caused by mechanical stresses during nebulization presents an additional challenge (Figure 2B), with the factors mediating phage susceptibility to degradation still poorly understood. Aerosolized phage therapy also suffers from similar limitations affecting the inhalation of other medications, one of the most prominent being the loss of medication through deposition in the upper airways (Figure 2C). Subsequently, only a fraction of an inhaled dose of medication will end up depositing at a site of infection in the lower respiratory tract [118,119]. Following inhalation and deposition, phages may also be removed from the body by the host immune system or through mucociliary clearance (Figure 2D) [120]. Unfortunately, the precise numbers of phage that are lost through these mechanisms during delivery in humans are not known; however, the high levels of variability across all aspects of aerosolized delivery may make this information difficult to establish. The various ways in which the number of viable phages may be reduced highlight the need to minimize titer loss in the early stages of delivery (formulation and aerosolization) to ensure high bactericidal activity for phages depositing at sites in the lower airways.

One of the barriers for the optimization of aerosolized phage delivery is a lack of standardized reporting of study details. Some studies do not report genomic and morphological characteristics of their aerosolized phages. The omission of this data makes it difficult to draw definitive conclusions across different studies, and with no known phage characteristic(s) currently linked to stability in aerosols, details on the phage types used in these experiments may be crucial to understanding which phages are best suited for aerosolization [79,80]. To aid in this, aerosolized phage studies should report the phage family as well as any genomic and morphological characteristics of phages used for experiments or treatment. This will facilitate dedicated studies of phages and their morphology to further elucidate potential links between phage types and aerosolization stability.

As the components of a liquid formulation will affect its capacity to be aerosolized, phage formulation ingredients must also be well reported [85,121,122]. Additionally, components of the aerosolization medium may influence phage stability during both storage and delivery [65,85,96]. Therefore, it is vital that studies examining aerosolized phage note the components of any buffers and aerosolization media used during manufacturing, storage, and delivery processes. Future studies systematically testing different stabilizing additives or compounds are warranted for the identification of components that protect phages from damage during nebulization.

Owing to the fact that different types of nebulization have been shown to have varying effects on phages, the reporting of the parameters during nebulization must be thorough [64,65]. Some studies omit the model of nebulizer used or even the type of nebulization used for delivery. Additionally, studies do not always report the nebulization parameters, such as operating frequencies, airflow rates and subsequent particle sizes and aerosol output. These factors may vary between different models and brands within each nebulizer type and affect phage viability differently. As there are currently no phage characteristics associated with titer loss following nebulization, the reporting of these nebulizer specifications may be crucial to understanding how phages are affected by different devices and operating parameters.

## 7. Conclusions

The aerosolization of phage in either solid or liquid formulations appears to be a promising method for delivery to sites of infection in the lower airways. All methods of aerosolization have been shown to result in a loss of infectious phage titer in addition to reduction as a part of manufacturing and storage processes. Studies have suggested that titer loss is phage specific, and little to no phage or formulation-specific characteristics have been associated with loss of titer following aerosolization. This remains a major hindrance to full clinical translation of inhaled phage therapy. Further investigation on the parameters and conditions for the manufacture, storage and delivery of aerosolized phage products will bridge this knowledge gap and inform the design of therapeutically effective aerosolized phage formulations. Future studies must identify suitable phages or phage characteristics, buffer conditions, and aerosolization conditions resulting in minimal titer loss, ensuring that aerosolized phage formulations have maximal therapeutic activity upon delivery.

## Figures and Tables

**Figure 1 cells-12-02057-f001:**
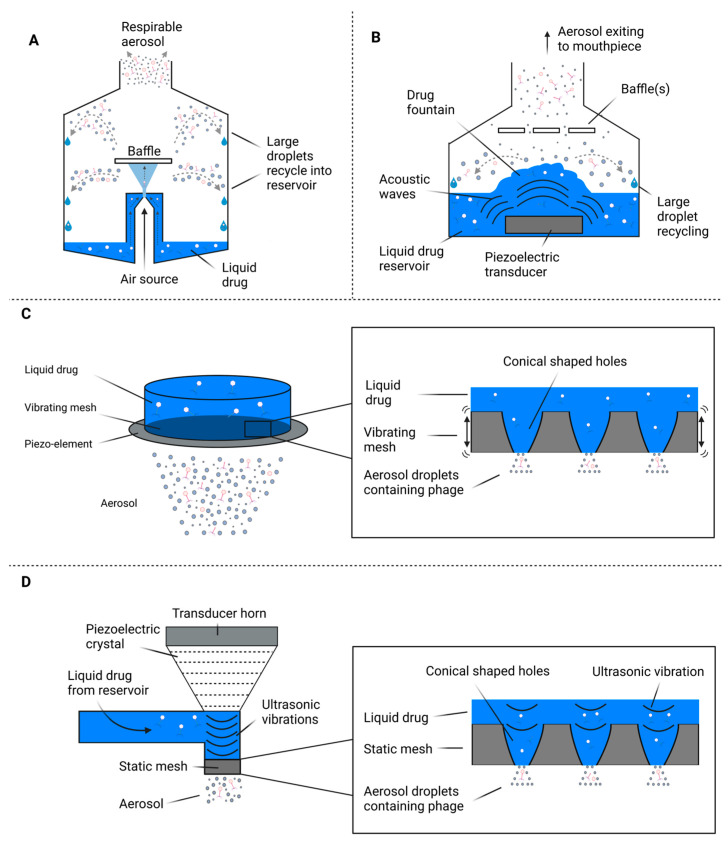
Devices used for liquid aerosolization of phages. Diagrams represent the general principle of phage delivery using: (**A**) Compression/jet nebulization, (**B**) Ultrasonic nebulization, (**C**) Vibrating mesh nebulization, (**D**) Static mesh nebulization.

**Figure 2 cells-12-02057-f002:**
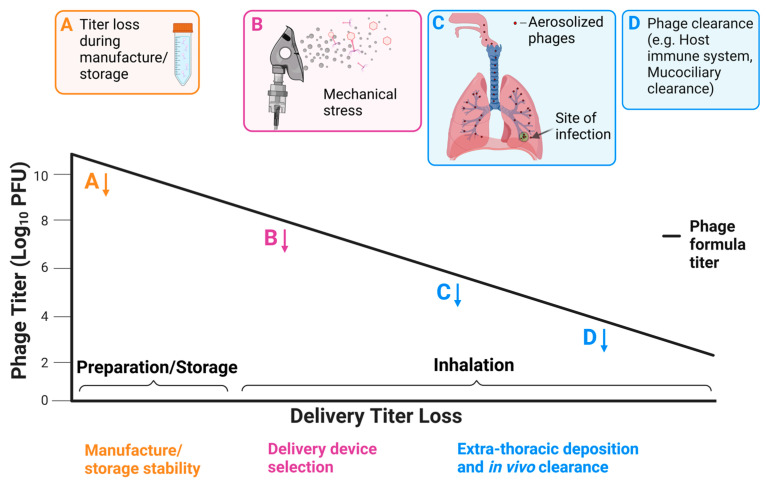
Timeline of aerosolized phage delivery to sites in the lower respiratory tract. Various points (**A**–**D**) along the timeline of aerosolized phage delivery highlight where phage volume and/or titer may be reduced; (**A**): Titer reduction due to instability during storage. (**B**): Mechanical stress destroys phage particles during nebulization. (**C**): Secondary deposition in the upper respiratory tract and in extra-thoracic regions. (**D**): In vivo phage clearance by the host, e.g., anti-phage immune response and mucociliary clearance. Y-axis is a demonstrative hypothetical representation of titer loss during aerosolization.

**Table 1 cells-12-02057-t001:** Studies examining phage viability using jet, vibrating and static mesh nebulizers.

Type of Nebulizer	Phage Viability (Log_10_ Reduction)	Bacterial Load	Particle Size (Mean Diameter)	Lung Deposition	Study [Reference]
Jet	1.3	N/A	3.23 µm	2.4% FPF	In vitro [64]
	N/A	N/A	N/A	12% in particles <4.7 µm	In vitro [70]
	0.15–2.45	N/A	N/A	0.2–2.8 log_10_ reduction at tracheal deposition	In vitro (6 phages) [81]
	3.7	N/A	N/A	N/A	In vitro [65]
	0.04, 1.02, 3.25	N/A	N/A	N/A	In vitro (3 phages) [82]
	N/A	N/A	4.98 µm	2.15 × 10^8^ PFU delivered 3.02 × 10^7^ PFU alveolar deposition (predicted)	In vitro [69]
	N/A	4 log_10_ CFU reduction after 3 days	N/A	>10^9^ PFU delivered ≈1 × 10_4_ PFU/g lung tissue	In vivo (Mouse) [83]
Vibrating mesh	0.5	N/A	5.3 µm	4.1% FPF	In vitro [64]
	N/A *	Reduced only in surviving animals (50%)	3.1 µm	2 × 10^10^ PFU delivered 1.4 × 10^6^ PFU/g lung tissue	In vivo (Rat) [53]
	0.4	N/A	N/A	N/A	In vitro [65]
	N/A	N/A	5.83 µm	2.15 × 10^8^ PFU delivered 2.96 × 10^7^ PFU alveolar deposition (predicted)	In vitro [69]
Static mesh	0.3	N/A	5.6 µm	15% FPF	In vitro [64]
	1.3	≈1.5 log_10_ CFU reduction after 21 h	1.13 µm	2 × 10^11^ PFU delivered 5.8 × 10^6^ PFU/g lung tissue	In vivo (Pig) [84]

* This study did not report the titer reduction in their phage; however, they noted a 93% phage viability following nebulization. N/A, not assessed; FPF, fine particle fraction; PFU, plaque-forming units; CFU, colony-forming units.

**Table 2 cells-12-02057-t002:** Overview of various case studies utilizing aerosolized phage delivery.

Condition + Pathogenic Organism (Reference)	Phage(s)	Delivery Method	Outcome
17-YO girl with CF + chronic *Achromobacter xylosoxidans* lung infection [99]	Two (siphovirus) phages from the Ellavia Institute	Jet Neb: 6 × 10^8^ PFU diluted in 2–3 mL 0.9% NaCl 1× daily for 20 days, alongside oral delivery 2× daily Treatment repeated at 1, 3, 6 and 12 months	Bacterial load reduction not measured Patient lung function significantly improved
88-YO man CRAB lung infection [114]	Ab-SZ3 phage (siphovirus)	VM Neb: 1× daily for 2 days in 0.9% NaCl, then every 12 h for 13 days Titer gradually increased throughout treatment (5 × 10^6^ to 5 × 10^10^ PFU)	Complete clearance of pathogen Patient lung function improved
52-YO patient with CRAB lung infection [115]	Single phage AbW4878Ø1	IV: 1 × 10^9^ PFU/mL diluted in 50 mL of 0.9% NaCl 2× daily for 14 days (Second round) IV and VM Neb: 0.1 × 10^9^ PFU/mL diluted in 10 mL of 0.9% NaCl 2× daily for 21 days	Complete clearance of pathogen Patient lung function significantly improved
12-YO CF individual with pan-drug-resistant *Achromobacter xylosoxidans* lung infection [113]	Cocktail 1: 3 siphovirus phages Cocktail 2: 4 siphovirus phages	VM Neb: 5 mL 4 × 10^10^ PFU/mL Cocktail 1 diluted tenfold in 0.9% NaCl 3× daily for 2 days (4 months post PT) VM Neb: 5 mL Cocktail 2, diluted tenfold in 0.9% NaCl 2× daily for 14 days, as well as 30 mL instilled in each pulmonary lobe	Patient lung function significantly improved BAL remained positive for *A. xylosoxidans* Pathogen eradicated 2 years after PT stopped
4 cases of CRAB lung infections in critical COVID-19 patients aged 62–81 [117]	Cocktail of phages ΦAb121 and ΦAb124 delivered to patients 1–4	VM Neb: 10^8^ PFU/mL in 10 mL 0.9% NaCl 2 doses with 1-h interval. Patient 2 also had phages delivered post PT topically via gauze pad for a jugular incision	Decrease in semi-quantitative bacterial load after PT for all patients Patient 3’s CRAB completely eliminated All patients showed improved clinical condition
41-YO patient with MDR *Pseudomonas aeruginosa* infection [116]	Single phage vFB297 (myovirus)	Neb: 5 × 10^9^ PFU daily for 5 days + 2 additional days. Additional 5 day course of phages after 465 days	Patient lung function significantly improved Sputum samples remained positive for *P. aeruginosa*

YO, year old; CF, cystic fibrosis; Neb, nebulization; PFU, plaque-forming units; CRAB, carbapenem-resistant Acinetobacter baumannii; VM, vibrating mesh; PT, phage therapy; BAL, bronchoalveolar lavage; COVID-19, coronavirus disease 2019.

## Data Availability

No new data were created or analyzed in this study. Data sharing is not applicable to this article.

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
