# Peer review of "Stability Considerations for Bacteriophages in Liquid Formulations Designed for Nebulization"

_cells, 2023, doi:10.3390/cells12162057_

Round 1
Reviewer 1 Report
Compliments for the explanatory figures and related text Figures and
Author Response
We thank the Reviewer for their comments and feedback on the review.
Reviewer 2 Report
The manuscript entitled “Stability considerations for bacteriophages in liquid formulations designed for nebulization” is review for bacteriophages in liquid formulations designed for nebulization.
Manuscript is written well but need to some modification to be published.
Authors need to provide briefly other treatment methods with pros and cons, and benefits of nebulization at introduction.
Authors need to provide references at line 103.
If possible, authors need to provide more challenging burden of phage aerosolization with reference at line 125.
If possible, authors need to provide the table for the comparison of 3 methods to deliver the drug to lung.
The figure has low resolution, authors need to increase that.
Sometime I feel confusion because of phage activity, I recommend authors need to carefully check and replace with phage number or viable phage number at some point.
Authors need to double check whether the reference written consistently for example, Journal name is abbreviated name or full name
Author Response
We thank the Reviewer for their feedback and comments on the review.
Please see the attachment for detailed responses to the comments.

Reviewer 3 Report
It is a review that addresses a controversial but potentially useful topic such as phage therapy. It discusses a lot of information derived from previous studies, clearly establishing advantages and disadvantages of the strategy. It establishes taking into account the stability of the phages in the preparations. Since it is a review, its objective is to gather the available information and discuss it. Since the stability of the phages in the preparations is critical, the manuscript provides which topics should be worked on to ensure their effectiveness. The conclusions are consistent with the evidence and arguments presented and they address the main question posed. The references are appropriate.
The manuscript is a valuable contribution to the theme of phage therapy and presents very well, with a good english, the advantages and disadvantages of this strategy. I have no observations about the content. Just recommend that the genus and species of microorganisms in italics write. See References.
Author Response
We thank the Reviewer for their kind feedback and comments on the review.
We also thank the Reviewer for the reminder to italicize the genus and species names in the references, which has now been corrected in the text.
